# Spatial-Temporal Characteristics in Grain Production and Its Influencing Factors in the Huang-Huai-Hai Plain from 1995 to 2018

**DOI:** 10.3390/ijerph17249193

**Published:** 2020-12-09

**Authors:** Chunshan Zhou, Rongrong Zhang, Xiaoju Ning, Zhicheng Zheng

**Affiliations:** 1School of Geography and Planning, Sun Yat-sen University, Guangzhou 510275, China; zhoucs@mail.sysu.edu.cn (C.Z.); zhangrr5@mail2.sysu.edu.cn (R.Z.); 2School of Resource and Environment, Henan University of Economics and Law, Zhengzhou 450000, China; 3School of Environment and Planning, Henan University, Kaifeng 475000, China; zzc3148@163.com

**Keywords:** grain production, spatial–temporal characteristic, influencing factors, the Huang-Huai-Hai Plain

## Abstract

The Huang-Huai-Hai Plain is the major crop-producing region in China. Based on the climate and socio-economic data from 1995 to 2018, we analyzed the spatial–temporal characteristics in grain production and its influencing factors by using exploratory spatial data analysis, a gravity center model, a spatial panel data model, and a geographically weighted regression model. The results indicated the following: (1) The grain production of eastern and southern areas was higher, while that of western and northern areas was lower; (2) The grain production center in the Huang-Huai-Hai Plain shifted from the southeast to northwest in Tai’an, and was distributed stably at the border between Jining and Tai’an; (3) The global spatial autocorrelation experienced a changing process of “decline–growth–decline”, and the area of hot and cold spots was gradually reduced and stabilized, which indicated that the polarization of grain production in local areas gradually weakened and the spatial difference gradually decreased in the Huang-Huai-Hai Plain; (4) The impact of socio-economic factors has been continuously enhanced while the role of climate factors in grain production has been gradually weakened. The ratio of the effective irrigated area, the amount of fertilizer applied per unit sown area, and the average per capita annual income of rural residents were conducive to the increase in grain production in the Huang-Huai-Hai Plain; however, the effect of the annual precipitation on grain production has become weaker. More importantly, the association between the three factors and grain production was found to be spatially heterogeneous at the local geographic level.

## 1. Introduction

China is an important food-producing country in the world, as well as a large food consumer. China’s food self-sufficiency rate has reached more than 95% [1]. Although the current grain supply and demand in China maintains a balance of total quantity and a surplus in harvest, the small per capita arable land area, low mechanization level, and family-based business units determine that the current arable land has an extremely limited potential for increasing grain production [2,3]. From 1850–1900 to 2006–2015, the mean land surface air temperature increased by 1.53 °C [4]. Climate change has already affected food security due to warming, changing precipitation patterns, and the greater frequency of some extreme events [4]. Under the dual constraints of climate change and the process of urbanization, unpredictable meteorological disasters, limited arable land resources, huge population pressure, and the diversified consumer demand of residents directly generated strong demands for stable grain production [5,6]. Additionally, the outbreak of COVID-19 in 2019 led to labor shortages and supply chain disruptions, which affected the food security of some countries and regions [7]. Certain countries have even banned the export of food, leading to large fluctuations in global food prices and casting a shadow over the world’s food crisis. Although China’s major food crops, such as rice and wheat, are less dependent on the international market and national food security will not be affected by the agricultural trade restrictions brought about by the COVID-19, ensuring China’s food security in the later period of the epidemic is of great strategic significance for domestic economic recovery and social stability.

The Huang-Huai-Hai Plain (HHH), located in the north of the country, is a high-yield agricultural region [8], accounting for 70% of wheat and 30% of maize production in China [9]. Because of its importance in grain production and China’s self-sufficiency in food, it is known as the ‘Breadbasket of China’ [10,11]. Changes in grain production in the HHH can have direct impacts on both the national economy and food security of China [12]. Therefore, it is of great significance to understand its temporal and spatial characteristics and the influencing factors of grain production in the Huang-Huai-Hai plain in order to ensure national food security.

Previous studies have pointed out that while grain production in the Huang-Huai-Hai Plain is increasing, the difference in grain yields in various regions is gradually shrinking, but the spatial distribution pattern of grain production in the southern region of the Huang-Huai-Hai Plain is higher than that in the northern region, which remains unchanged [13,14,15]. Later studies attribute the distribution characteristics of grain production in the HHH to socio-economic factors [13,14,16]. The increase in grain production in the HHH is mainly due to the improvements of crop varieties, fertilizers and effective irrigation area. For example, the amount of fertilizer used has increased by about 400% and the effective irrigation area has increased by about 20% [10]. Liu, Tang [15] used the spatial lag model to reveal the factors affecting the differentiation in grain yield in the HHH from 1995 to 2010, and found that farmers’ per capita net income, effective irrigation area ratio, and industrial structure had significant positive effects on grain yield. At the same time, studies have also found that although grain production and fertilizer input in the HHH are still significantly positively correlated, the current application of fertilizers in actual production is extremely unbalanced and unreasonable [17]. It is essentially important to improve fertilizer use efficiency in order to save resources as well as increase yield [10].

A number of studies have also emphasized the role of climate factors, especially precipitation, in grain production in the HHH [18,19,20,21,22]. Qu, Li [20] indicated that the increase in precipitation in the HHH can significantly increase the food production in the HHH, but the increase in thermal resources will increase the shortage of water resources and offset the impact of the increase in temperature. Xiao, Qi [23] reveled that climate change reduced the potential winter wheat yield of 80% of the stations by 2.3–58.8 kg∙yr^−1^, while at the same time it is pointed out that increasing the heat time of the wheat growth period is essential to alleviate the impact of the shortening of the growth period caused by warming climatic conditions. However, if the advancement of agricultural technology and other non-climatic factors are taken into account, for every 1 °C increase in the average temperature of the Huang-Huai-Hai Plain, the winter wheat yield in the north will increase by 2.1%, and the yield in the south will decrease by 4.0% [22].

Therefore, most of the previous research on grain production in the Huang-Huai-Hai Plain mainly focuses on the factors affecting grain production, that is, mainly from two aspects: socio-economic factors and climatic factors. Further, these studies are only carried out on one level, which separates the comprehensive impacts of climate and socio-economic factors on grain production. This will inevitably affect the final assessment results. To complete these data, this paper aims to explore the influencing factors of grain production in HHH by incorporating the climate factors and socio-economic factors into the models so as to provide a reference for ensuring food security and relevant departments to make decisions.

## 2. Study Area and Data Sources

### 2.1. Study Area

The Huang-Huai-Hai Plain (HHH), the second largest plain in China, is located at 32°–40° N and 114°–121° E, with a land area of about 4 × 10^5^ km^2^, spanning seven provinces and cities of Beijing, Tianjin, Hebei, Shandong, Henan, Anhui, and Jiangsu (Figure 1). The HHH belongs to the warm temperate semi-humid monsoon climate zone and is one of the most sensitive areas to climate change in China. The winter climate in this area is typically dry and cold, spring is dry, with less rain and much evaporation, and summer is characterized by high temperatures and heavy rainfall, including high intense rainfall that often leads to summer floods [24]. As an important agricultural region, this area has a long history of farming and is an important grain production base for food security in China, with its sown area of 20.4% of the nation’s farmland and 23.6% of the whole nation’s grain yield [25].

Here, wheat and maize occupy a larger proportion in the structure of grain production. The annual double-cropping system of winter wheat and summer maize is the most popular planting pattern. In addition, the yield of wheat and maize accounts for approximately 61% and 31% of the total national output, respectively [26]. Therefore, it is necessary to determine the pattern change rules and influencing factors of grain production in upgrading HHH grain production and ensuring national food security.

### 2.2. Data Sources and Processing

The historical socio-economic data used in this paper were collected from the Statistical Yearbook published by the National Bureau of Statistics of China, which includes annual data on the grain yield per unit area, the sown area of grain crops, the amount of agriculture fertilizer application, the amount of effective irrigation area, the amount of pesticide application, the amount of mechanical power, and the per capita annual income of rural residents in the HHH from 1995 to 2018. According to China’s statistics, grain production includes corn, rice, soybeans, wheat, potatoes, and sweet potatoes.

The historical climate data were collected from the Chinese meteorological data hub (https://data.cma.cn), including the annual precipitation, annual temperature, and annual sunshine duration of 123 meteorological stations in the HHH from 1995 to 2018. The meteorological data need to be processed by SQL-Server (Microsoft Corporation, Redmond, WA, USA) and ArcGIS10.2 software (Environmental Systems Research Institute Inc, Redlands, CA, USA), which can analyze the average annual temperature, annual precipitation, and average annual sunshine hours of each city in different years.

Grain production is influenced by a variety of natural and socio-economic factors. In this paper, based on the previous literature [27,28,29], the grain yield per unit sown area (kg/km^2^) was taken as the dependent variable, and climate and socio-economic factors were taken as the independent variables. Among them, the climate indexes included the annual average temperature (°C), the annual average precipitation (mm), and the annual average sunshine duration (h), and the social-economic factors included the proportion of effective irrigation area (%), amount of fertilizer application per unit sown area (t/km^2^), amount of pesticide application per unit sown area (t/km^2^), amount of mechanical power per sown area (kw/km^2^), and per capita annual income of rural residents (RMB). Table 1 summarizes the data used in this study.

## 3. Methods

### 3.1. Exploratory Spatial Data Analysis (ESDA)

Exploratory spatial data analysis is a collection of techniques for describing and visualizing spatial distributions, determining atypical locations or spatial outliers, discovering spatial associations, clusters, or hot spots, and to infer spatial characteristics or other forms of space heterogeneity [30]. In general, global and local spatial autocorrelation (or hot spots analysis) is often used to explore the spatial characteristics of observations [31].

#### 3.1.1. Global Spatial Autocorrelation

Global spatial autocorrelation is used to test the spatial correlation of the observations of spatial units within the study area [32], and is mainly measured by the Global Moran’s I, which was first proposed by Moran [33]. The Moran’s I can be calculated using Equation (1):(1)$$I=\frac{n\sum_{i=1}^{n}\sum_{j=1}^{n}w_{ij}\left(x_{i}-\bar{x}\right)\left(x_{j}-\bar{x}\right)}{\sum_{i=1}^{n}\sum_{j=1}^{n}w_{ij}\sum_{i=1}^{n}{\left(x_{i}-\bar{x}\right)}^{2}}$$
where *I* represents Moran’s *I*, *n* is the number of spatial units (in this study, *n* = 59), *x_i_* and *x_j_* are the observations of spatial units *I* and *j*, respectively, $\bar{x}$ is the average value of observations of spatial units, and wij is the spatial weight matrix, where *w_ij_* = 1 if spatial units *I* and *j* share a common border and *w_ij_* = 0 otherwise. The values of Global Moran’s *I* range from −1 to 1. If *I* < 0, it means there is a negative spatial correlation in the space, while if *I* > 0, it means there is a positive spatial correlation, an d if *I* = 0, it means there is no spatial correlation.

The significance of Moran’s *I* is usually measured by *Z* statistics using Equation (2):(2)$$Z\left(I\right)=\frac{I-E\left(I\right)}{\sqrt{Var\left(I\right)}}$$
where *E*(*I*) and *Var*(*I*) are the expected value and variance of Moran’s *I*, respectively.

#### 3.1.2. Hot Spot (Getis-Ord *G_i_*^*^) Analysis

The Getis-Ord *G_i_*^*^ is commonly used for hot spot analysis, which can identify clustering relationships at different spatial locations. Compared with the local spatial autocorrelation, the Getis-Ord *G_i_*^*^ is more sensitive to the identification of cold and hot spots, and can fully reflect the high or low value distribution relationship between a certain geographic element and other surrounding elements [34]. The formula is [34,35,36,37]:(3)$$G_{i}{}^{*}=\frac{\sum_{j=1}^{n}w_{ij}\left(x_{j}-\bar{x}\right)}{\sqrt{\frac{1}{n}\sum_{j=1}^{n}x_{j}{}^{2}-{\bar{x}}^{2}\;\cdot }\sqrt{\frac{n}{n-1}\sum_{j=1}^{n}w_{ij}{}^{2}-\frac{n}{n-1}{\left(\sum_{j=1}^{n}w_{ij}\right)}^{2}}}$$
where $\bar{x}=\frac{1}{n}\overset{n}{\underset{j=1}{\sum }}x_{j}$, *n* is the number of spatial units (in this study, *n* = 59), and *w_ij_* is the spatial weight matrix, where *w_ij_* = 1 if spatial units *I* and *j* share a common border and *w_ij_* = 0 otherwise.

The significance of *G_i_*^*^ is usually measured by *Z* statistics using Equation (4):(4)$$Z\left(G_{i}{}^{*}\right)=\frac{\left[G_{i}{}^{*}-E\left(G_{i}{}^{*}\right)\right]}{\sqrt{Var\left(G_{i}{}^{*}\right)}}$$
where *E*(*G_i_*^*^) and Var(*G_i_*^*^) are the expected value and variance of *G_i_*^*^, respectively. If *Z*(*Gi*^*^) is significantly positive, it indicates that the observations around the spatial unit *i* are relatively high (higher than the average), and are high-value clusters in the space, belonging to hot spots; on the contrary, if *Z*(*G_i_*^*^) is significantly negative, it indicates that the observations around the spatial unit *i* are relatively low (lower than the mean), and are low-value clusters in the space, belonging to cold spots. The larger (or smaller) the *Z*(*G_i_*^*^) is, the more intense the clustering of high (or low) values. A *Z*(*G_i_*^*^) near zero indicates no apparent spatial clustering.

### 3.2. Gravity Center Model

The gravity center model is used to measure the overall distribution of a certain attribute in a region. It can provide a concise and accurate feature of the distribution of the attribute in the space, and can indicate the general trend and central location of its distribution. We assumed that a large region (such as an administrative region) consists of several subregions, and so the gravity center of grain production in the region could be calculated by the grain production and geographic coordinates of each sub-region. The formula is [38]:(5)$$X=\frac{\sum_{i=1}^{n}M_{i}X_{i}}{\sum_{i=1}^{n}M_{i}};\;Y=\frac{\sum_{i=1}^{n}M_{i}Y_{i}}{\sum_{i=1}^{n}M_{i}}$$

In Equation (2), *X_i_* and *Y_i_* represent the geographic coordinates of the ith subregion. *M_i_* represents the grain yield per unit sown area of the subregion. *X* and *Y* represent the gravity center of grain production in a large region. Using formula (3), the moving distance of the gravity center in grain production can be obtained, which can reflect the evolution of the gravity center of a property in a region;
(6)$$D_{ij}=R\times \sqrt{{\left(X_{i}-X_{j}\right)}^{2}+{\left(Y_{i}-Y_{j}\right)}^{2}}$$

In Equation (3), *D_ij_* is the gravity center movement distance (km) of grain production from *j* to *i* years. (*X_i_*, *Y_i_*) and (*X_j_*, *Y_j_*) are the gravity center coordinates of grain production in the *i* and *j* years. R is typically 111.111, which represents the coefficient of the spherical longitude and latitude coordinates converted to plane distance.

### 3.3. Spatial Panel Data Model

When the data have spatial autocorrelation effects, the residuals are no longer independent of each other; thus, it is not appropriate to use the ordinary least square regression (OLS) model. Instead, the spatial lag model (SLM) or the spatial error model (SEM) should be used for analysis. The formula is [39,40,41]:(7)$$SLM:\;\;\;\;y=\rho w_{ij}y+x\beta +\mu$$
(8)$$SEM:\;\;\;\;\{\begin{array}{c} y=x\beta +\varepsilon \\ \varepsilon =\lambda w_{ij}+\mu \end{array}$$

In Equations (4) and (5), *y* is the dependent variable, *x* is the explanatory variables, *W_ij_* is the space weight matrix, *ρ* is the spatial hysteresis parameter, *β* is the parameter vector, *μ* is the random interference term, *ε* is the regression residual vector, and *λ* is the autoregression parameter.

### 3.4. Geographically Weighted Regression (GWR) Model

Geographically weighted regression models are superior to traditional regression models such as ordinary least squares (OLS). The geographical weighted regression (GWR) model can fully consider the spatial characteristics of each influencing factor, and more accurately show the spatial relationship between independent and dependent variables [42]. The form of a GWR model is as follows:(9)$$Y_{i}=\beta_{0}\left(u_{i},v_{i}\right)+\sum_{\lambda }^{n}\beta_{\lambda }\left(u_{i},v_{i}\right)X_{i\lambda }+\varepsilon$$

In Equation (6), Y*_i_* represents the grain production in region *i*, *β*_0*(ui,vi)*_ represents a constant, *β*_λ*(ui,vi)*_ represents the regression coefficient, *(u_i_*,*v_i_)* represents the geographic location of the cities *i*, *X_iλ_* represents the parameter value of the *λ* independent variable of city *i*, and *ε* represents the random error. The optimal bandwidth distance can be obtained automatically in GWR4.0 corrected by finite correction of the Akaike Information Criterion (AICc). The smaller the AICc value, the higher the goodness of fit of the model will be [43].

## 4. Results

### 4.1. Temporal Changes of Grain Production in the HHH

From 1995 to 2018, the grain yield per unit of sown area in the HHH has shown a steady increase (Figure 2), which can be divided into three stages, as follows: the fluctuating growth stage (1995–2005), the steady growth stage (2005–2015), and the slow descent stage (2015–2018). Firstly, in the fluctuating growth stage (1995–2005), although the grain yield per unit sown area has increased in this stage, the fluctuation range is relatively large and the grain yield per unit of sown area gradually stabilized in 2005. Secondly, during the steady growth stage (2005–2015), the grain yield per unit of sown area showed a characteristic of small fluctuation growth. Finally, a slow decline began to appear in the grain yield per unit of sown area in the HHH during the slow descent stage (2015–2018).

### 4.2. Spatial Characteristics of Grain Production in the HHH

We classified the grain yield data of each urban unit of sown area into four types according to 2000–3500, 3500–5000, 5000–6500, and 6500–8000 kg/km^2^ using ArcGIS10.2 software. For the convenience of analysis, four time sections of 1995, 2005, 2015, and 2018 were selected for study, for which the spatial distribution characteristics of the grain production pattern were discussed (Figure 3).

It can be seen from Figure 3 that the spatial variation in grain yield per unit of sown area in each city is very significant, and the overall grain yield shows an increasing trend. Specifically, the number of cities in the HHH where the grain yield per unit of sown area remained within the range of 2000–3500 kg/km^2^ continued to decrease, and was mainly distributed in Zhangjiakou, Cangzhou, Zhengzhou, Luoyang, Nanyang, Sanmenxia, and Bozhou in 1995, Zhangjiakou and Sanmenxia in 2005, and only in Zhangjiakou in 2015. In 2018, the number of cities of this type decreased to zero.

The number of cities within the range of grain yield per unit of sown area of 3500 to 5000 kg/km^2^ also showed a downward trend, and the type area shrank from 30 cities in 1995 to 5 cities in 2018, and presented a layout trend of agglomeration to dispersion in spatial distribution.

The number of cities in the range of 5000–6500 kg/km^2^ showed a rising–falling–rising trend. Among them, the number of cities in this type of area increased from 19 to 35 in 1995–2005, reduced from 35 to 28 in 2005–2015, and increased from 28 to 40 in 2015–2018. The spatial distribution of this type overall showed a tendency varying between scattered and clustered development, and appeared roughly from east to west, and from south to north.

The trend of unit of area sown to grain production maintained in the 6500–8000 kg/km^2^ interval of urban change was different from the above three kinds. In 1995, the number of cities within this range was 3, which increased to 5 in 2005 and 24 in 2015, and reduced to 14 in 2018, which reflects a characteristic of a sharp increase followed by a slow decline. However, they were still mainly distributed in the central and southern parts of the HHH.

In terms of spatial distribution, the areas with high grain yield per unit of sown area in the Huang-Huia-Hai Plain were mainly distributed in the east and south, while the areas with low grain yield per unit of sown area were mainly distributed in the west and north of the HHH, which indicated that the grain production capacity of the eastern and southern regions of the HHH was higher, while that of the northern and western regions was lower.

### 4.3. Dynamic Change of Barycenter of Grain Production in the HHH

Figure 3 reflects the static distribution pattern of grain production in the HHH in 1995, 2005, 2015, and 2018, but does not reflect the dynamic change trend. Therefore, we used Equation (2) to calculate the barycenter of grain production in the HHH from 1995 to 2018 (Figure 4), and then used Equation (3) to calculate the barycenter movement distance (Table 2) to analyze the dynamic change characteristics of the grain production pattern.

According to Figure 4, the grain production center in the HHH generally shifted from southeast to northwest in Tai’an, and gradually stabilized in the central area of the HHH with the passage of time. Specifically, from 1995 to 1997, the grain production center of HHH was distributed in Jining city, which moved first to the northwest and then to the southwest. From 1998 to 2000, the movement direction of the barycenter in grain production remained highly stable; that is, it continued to move in the northwest direction. The barycenter of grain production began to enter Tai’an City. From 2000 to 2001, the movement direction of the barycenter in grain production was reversed and shifted to the southeast.

In 2001–2003, the barycenter of grain production shifted first to the southwest and then to the northeast, and in 2003–2005, it shifted first to the southwest and then to the southeast. From 2005 to 2011, the barycenter of grain production fluctuated in all directions. From 2011 to 2013, the grain production center assumed a similar change trend to that from 2001 to 2003. From 2013 to 2015, the grain production center first moved to the northeast and then to the northwest.

From 2015 to 2018, the grain production center of gravity showed characteristics of moving from southeast to northwest, but it was still stable in the territory of Tai’an City, that is, the southeast of the HHH. The dynamic shift in the grain production center in the HHH indicates that the regional grain production capacity has the characteristics of non-stationarity in time and non-equilibrium in space at the same time, and the shift in the grain production center from southeast to northwest indicates that the grain production capacities in the western and northern parts of the HHH were continuously enhanced. In addition, the grain production center in the HHH tended to be stable over time, and it was concentrated in the border area between Jining and Tai’an.

Table 2 shows that the barycenter of grain production in the HHH as a whole moved from the southeast to northwest from 1995 to 2005, with a distance of 17.7 km. From 2005 to 2018, the barycenter of grain production moved to the northwest with a distance of 9.4 km, which was significantly smaller than that from 1995 to 2005, which confirmed that the barycenter of grain production in the HHH showed good time-stability characteristics over time; however, this could not cover up the disequilibrium in the spatial characteristics of grain production. On the whole, from 1995 to 2018, the center of gravity of grain production moved 26.1 km to the northwest. The center was still stable in the territory of Tai’an city, that is, to the southeast of the HHH.

### 4.4. Spatial Correlation Characteristics of Grain Production Pattern in the HHH

#### 4.4.1. Global Spatial Correlation Characteristics

Based on the grain yield data per unit sown area, the Moran’s *I* value, *Z* statistic, and *P*-value were calculated using Geoda software, and the spatial correlation characteristics of grain production are shown in Table 3.

It can be seen from Table 3 that the Moran’s *I* values from 1995 to 2018 were all greater than 0 and significant at the threshold level of 5%, indicating that the grain yield per unit sown area in the HHH was not randomly distributed but positively correlated. This indicates that the grain production layout showed strong spatial clustering characteristics. From 1995 to 1997, the Moran’s *I* value decreased from 0.4114 to 0.1718. This indicates that the agglomeration and development trend in grain production in the HHH weakened during this period. From 1997 to 2002, the Moran’s *I* value showed a fluctuating trend, ranging from 0.1718 to 0.3356. In this period, the grain production experienced a process of both agglomeration and dispersion development, but the change trend was small.

From 2002 to 2006, the Moran’s *I* value showed “up–down” fluctuation characteristics twice, and the change trend was more intense. This indicates that the grain production pattern of the HHH was greatly changed during this period. From 2007 to 2011, the increase in Moran’s *I* value indicated that the grain production distribution in the HHH was in an increasingly concentrated state in this period. From 2011 to 2012, the Moran’s *I* value changed from 0.2897 to 0.2401, a decrease by 0.0496 units, indicating that the clustering characteristics of grain production layout weakened during this period.

The Moran’s I value showed a continuous rising trend, indicating that the clustering characteristics of grain production distribution were enhanced from 2012 to 2014. The value of Moran’s *I* changed from 0.3727 to 0.2315 in 2014–2018, indicating that the agglomeration characteristics of grain production distribution during this period weakened. On the whole, the Moran’s *I* value experienced a change process of “down–up–down” in this period; however, the agglomeration and distribution characteristics of grain production did not change.

#### 4.4.2. Local Spatial Correlation Characteristics

The evaluation of the global spatial correlation feature has the defect of ignoring the instability of local spatial processes. Therefore, the local spatial correlation characteristics of grain production in the HHH can be analyzed by observing the *G_i_*^*^ index in 1995, 2005, 2015, and 2018 (Figure 5).

According to Figure 5, the numbers of hot spots, sub-hot spots, sub-cold spots and cold spots were 13, 21, 21, and 4, respectively, in 1995, and the cold spots were mainly distributed in Sanmenxia, Luoyang, and Nanyang. The sub-cold spots were distributed in Beijing, Zhangjiakou, Chengde, Baoding, Jiaozuo, Xinxiang, Zhengzhou, Kaifeng, Zhoukou, Zhumadian, Xinyang, Fuyang, Bengbu, Huainan, and Bozhou. Sub-hot spots were concentrated in Tianjin, Qinhuangdao, Langfang, Cangzhou, Hengshui, Xingtai, Shijiazhuang, Dezhou, Binzhou, Dongying, Heze, Anyang, Hebi, and Puyang. Hot spots were distributed in the eastern part of the HHH. In particular, Yantai, Qingdao, Weifang, Jinan, Rizhao, Linyi, Zaozhuang, Jining, Lianyungang and Xuzhou were the most typical ones.

In 2005, the numbers of cities in the four categories were 16, 14, 16, and 13. Compared with 1995, the number of cities in hot spots and cold spots increased, indicating that the agglomeration of grain production increased from 1995 to 2005. The cold spots were concentrated in the north and southwest of the HHH. The sub-cold spots were distributed in Qinhuangdao, Tangshan, Tianjin, Langfang, Baoding, Xinyang, Fuyang, Zhoukou, Xuchang, Zhengzhou, and Jiaozuo. The sub-hot spots continued to shrink, and were mainly distributed in Shijiazhuang, Cangzhou, and Hengshui. The spatial distribution of hot spots is clear, and the main distribution was in the middle and eastern part of the HHH. In 2015, the numbers of hot spots, sub-hot spots, sub-cold spots, and cold spots were 13, 20, 14, and 12 cities, respectively.

Compared with 2005, the number of cities in hot spots and cold spots decreased by 3 and 1, respectively, indicating that the local spatial clustering characteristics of grain production in the HHH were weakened from 2005 to 2015. In 2018, the hot spots, sub-hot spots, sub-cold spots, and cold spots included 10, 17, 22, and 10 cities, respectively. Compared with 2015, the number of cities in hot spots and cold spots decreased by three and two, and the number of cities in sub-hot spots increased by eight. This indicates that the local spatial agglomeration characteristics of grain production in the Huang-Huai-Hai Plain gradually weakened from 2005. In general, hot spots spread from the east of the HHH to the southeast and the central region, and sub-hot spots were mainly distributed in the central region. The cold spots and sub-cold spots were mainly distributed in the north and south regions.

### 4.5. Influencing Factors for the Changes of Grain Production in the HHH

#### 4.5.1. Analysis of the Spatial Spillover Effect of the Influencing Factors

First, SPSS software was used to eliminate the collinearity of variables, and finally four indexes were extracted, namely, the effective irrigation area (EIA), amount of fertilizer application per unit of sown area (AFA), per capita annual income of rural residents (PCI), and annual average precipitation (PRE). In addition, 3.3 proves that the grain production pattern in the HHH has dependent characteristics; therefore, the influence of the spatial spillover effect cannot be ignored. Secondly, the GeoDa software was used to obtain the parameter estimation results of the spatial lag model, spatial error model, and OLS model. Finally, the statistical values of LMLAG and R-LMLAG in the OLS results were significantly higher than those of LMERR and R-LMERR at the 10% level; as such, it was appropriate to use the spatial lag model to explore the key factors affecting grain production (Table 4).

According to Table 4, the coefficients of EIA were 0.505, 0.415, 0.532, and 0.588 in 1995, 2005, 2015, and 2018, respectively, and were effective at the 1% threshold level. This indicates that there was a positive correlation between the EIA and grain yield; that is, an increase in the EIA will bring about an increase in the grain yield. At the same time, the coefficient of EIA on the whole was on the rise, indicating that it has a stronger positive effect on grain yield. The continuous improvement of irrigation and water conservancy facilities in the HHH is the reason for this phenomenon.

The coefficients of fertilizer application per unit of sown area in 1995 and 2005 were 0.008 and 0.208, respectively, which were both significant at the 1% threshold level, and the coefficients in 2015 and 2018 were 0.124 and 0.023, respectively, and were significant at the 5% threshold level. This shows that the positive effect of chemical fertilizer application per unit of sown area on grain production experienced a changing process of decline after rising first, reflecting that the increase in chemical fertilizer application per sown area from 1995 to 2005 significantly increased the grain production in the HHH.

From 2005 to 2018, the boosting effect on grain production was alleviated. The reason for this phenomenon may be that the increase in the use of chemical fertilizers in the low-level agricultural development stage has a significant effect on increasing grain production. As the amount of fertilizer input remains at a high level, the effect of increasing the amount of fertilizer input in the future is limited as regards improving the grain yield. The coefficients of rural residents’ per capita annual income in the four time sections were 0.405, 0.170, 0.124, and 0.050, which passed the significance tests at the critical value levels of 1%, 5%, 10%, and 10%, respectively.

The increase in per capita annual income continued to weaken the boost to grain production. The reason is that agriculture has been the main source of income for Chinese farmers to maintain their livelihoods for a long time. As farmers’ income levels increase, they have more funds to purchase agricultural machinery, fertilizers, pesticides, seeds, and other production materials, thereby achieving the goal of increasing grain yield. With the rapid advancement of urbanization, farmers’ income channels have become increasingly diversified, which has greatly reduced their dependence on agriculture, and the tendency of farmers to invest in non-agriculture has become more obvious. Therefore, to some extent, the increase in PCI has an inhibitory effect on grain output.

The coefficient of PRE was 0.508 in 1995 and passed the significance test of the 1% critical value level, indicating that the increase in PRE in that year played a promoting role in grain production. The coefficients of PRE were 0.016 and 0.148 in 2005 and 2015, both of which failed to pass the 10% significance test, indicating that the PRE increase did not have an obvious positive promotion effect on grain production. The underlying reason may be that with the continuous improvement of irrigation facilities, the impact of changes in PRE on grain production became weaker.

#### 4.5.2. Analysis of Spatial Heterogeneity of Influencing Factors

Based on the above research findings, the EIA, AFA, and PCI had a significant impact on the grain production in the HHH. However, SLM cannot explain the specific degree and scope of the impact of these three factors in space, and so a GWR model was explored to investigate the spatial difference in the influence of these three factors. The three factors—EIA, AFA, and PCI—were put into the model according to the criteria of AICc minimization.

The spatial distribution of Local R-squared values derived from the GWR model is displayed in Figure 6. The GWR results show that three factors explain 81.1% of the variance in grain production. Geographic variations in these factors describe a difference in the combined statistical influence of the three variables on grain production across cities in the HHH, from a very weak relationship (near 0.30) to a strong relationship (>0.80). We found that 50.94% of cities maintained local R-squared values of more than 50%. The predictive power of the model shows characteristics of increasing from east to west. The local R-squared map suggests that the predictive power of the analysis was greatest in relation to the northern part (Chengde, Baoding, and Zhangjiakou) and the southwestern part (Jiaozuo, Jiyuan, and Sanmenxia). The lower R-squared values demonstrate a poorer regression fit in the eastern parts of the HHH, such as Weihai, Yantai, and Qingdao.

To explore the strength of the influence of each of the three factors on grain production, we created maps for each factor, which represent the geographic distribution of their regression coefficient values across HHH, according to the results of the GWR modeling. The mapped regression coefficients are divided into five classifications through Natural Breaks. In Figure 7, white is used to indicate cities without data, and gradient shading is used to show cities with a significant relationship between variables and grain production.

The proportion of the effective irrigated area had a positive impact on grain production in the HHH, and its impact intensity presents a spatial distribution characteristic of “low west and high east”. A total of 75.47% of the cities showed a significant positive relationship between the effective irrigate area and production, mainly in Hebei and Henan provinces. The reason may be that these two provinces are mostly inland, with relatively drier climate and less rainfall; therefore, irrigation is mainly used to supply the water requirement of crops.

AFA had a positive effect on 94.34% of the cities in the HHH, which can significantly increase the food production of 39.62% of the cities, mainly in the central, northern, and southern regions of the HHH, such as Zhangjiakou, Puyang, and Fuyang. AFA had a negative effect on 5.66% of the cities, mainly in the eastern part of the HHH, such as Qingdao and Weifang. The impact of AFA on grain production is limited. In particular, with the long-term investment of chemical fertilizers by Chinese farmers on cultivated land, the impact of various chemical fertilizers applied by farmers on soil fertility has become nearly saturated. The majority of the chemical fertilizers play a role in maintaining soil fertility after application, so even if the input of chemical fertilizers is increased, the positive effect on crop yield is not clear enough.

The per capita income of rural residents had a positive impact on 86.79% of the cities in the HHH, of which only 9.43% passed the significance test, mainly in the northeastern part of the HHH, such as Chengde, Qinhuangdao, Tangshan, Langfang, and Cangzhou. The per capita income of rural residents had a negative impact on 13.21% of cities, and was concentrated in the southwestern region of HHH, that is, the western and southern regions of Henan Province, such as Sanmenxia, Nanyang, and Xinyang. As rural residents flow into developed urban areas, the non-agricultural income they earn from moving into cities has gradually become an important source of livelihood, and their dependence and emphasis on agriculture has gradually decreased; abandonment can even occur, which causes a huge negative impact on grain production.

## 5. Discussions 

Even our results suggested that the effect of precipitation on grain production became weaker, we should also be aware of the relationship between rainfall and groundwater; that is, rainfall becomes groundwater through infiltration, providing sufficient water for irrigating crops. With the improvement in irrigation facilities, irrigation has gradually become an indispensable and important means for stable grain production. This may also be the reason why the direct impact of PRE on grain production is gradually weakening and why the EIA is gradually increasing. Future research should focus on related research in this area. The reasons for the insignificant effect of temperature and sunshine duration may be attributed to two points: first, compared with precipitation, the heat resources of the HHH can well meet the growth needs of winter wheat and summer corn, and the yield is less affected by temperature and sunshine duration. At the same time, an increase in temperature and sunshine duration may increase evaporation, thereby offsetting the effect of precipitation. It is also possible to attribute this effect to precipitation rather than temperature and sunshine duration. Second, it may be that the crops in the research area are not subdivided. Different crops have inconsistent requirements for temperature and sunshine duration, which may weaken the effects of temperature and sunshine duration. This will also be the focus of future research. China’s food self-sufficiency rate has reached more than 95%. The concept of “with grain in the hand, the heart is unharried” has been made reality. However, in the face of the impact of global warming and COVID-19, as well as the requirements of new urbanization, as a country with one of the largest populations in the world, ensuring the stability of China’s food supply is not only related to national security, but is also related to the stability of the world.

Therefore, based on the results of this paper, the following policy suggestions are put forward to increase grain production in the HHH, an important grain production base for food security in China to maintain China’s food security. The suggestions include the following: to increase the construction investment for basic farmland infrastructures, such as irrigation facilities; to cultivate and promote good varieties and treatments, and implement soil testing and formula fertilization; to standardize the rural land market; to promote the transfer of rural land in an orderly manner; and to realize the large-scale management of cultivated land.

However, the impact of relevant agricultural policies issued by the country also needs to be considered in the future. From 1995 to 2018, the overall increase in grain production in HHH and the gradual reduction in spatial differences well reflects the background and national policies of the country in different periods. Since 1995, with the advancement of agricultural technology, the agricultural development of the HHH has been weakened by natural conditions, and the grain production has increased significantly [13]. At the same time, due to the popularization of fertilizers, pesticides, and irrigation, the gap in grain production in various regions in HHH is also narrowing.

Due to rapid urbanization, the conversion of fertile irrigated land to non-agricultural land seems to pose a potential threat to the food security of the HHH, and even to the whole of China [10]. Moreover, farmers are gradually migrating to cities in search of higher incomes due to the urban–rural development gap [44], which has caused the arable land in the HHH to be abandoned, resulting in the expansion of spatial differences in grain production in different regions in the HHH, and also threatening national food security. In order to ensure national food security, the central government proposed the construction of “high-standard basic farmland projects” and “agricultural modernization” to promote the large-scale, intensive, and modernized management of arable land in the HHH, thereby increasing the grain production of the HHH, and promoting the development of sustainable agriculture.

At the same time, we should not ignore the resistance, created by resource endowments, to sustainable agriculture in the HHH. In particular, the precipitation cannot meet the water demand of the crops [8,45], and water shortage is one of the major factors threatening the high and stable production of wheat [20]. The water consumption greatly exceeds the precipitation, and groundwater must be extracted to make up for the deficiency so as to maintain high yields [21]. Some areas in the HHH even appear salinized due to unreasonable irrigation [10], which poses huge challenges for the minimization of environmental impacts and the development of sustainable agriculture [46,47]. Therefore, the government must not only build irrigation facilities, but more importantly, must promote water-saving irrigation technologies, improve water resource utilization efficiency, implement drip irrigation and sprinkler irrigation [46], etc., so as to achieve sustainable agricultural production in the HHH. At the same time, in the long run, in the face of the encroachment of arable land in the promotion of urbanization, the amount of arable land in the future will also face severe challenges [10]. Promoting intensive agricultural production and improving the level of intensive use of agricultural production are also of great significance to ensure future food production [48]. Moreover, these actions can also enhance the ability to respond to natural disasters in the future. However, a sustainable food and agriculture system is one which is environmentally sound, economically viable, socially responsible, nonexploitative, and which serves as the foundation for future generations [49,50,51]. With the long-term development of intensive agriculture production in the Huang-Huai-Hai Plain, agricultural practices ranging from the development of irrigation projects to the use of agrichemicals have often had negative environmental impacts, such as wildlife kills, pesticide residues in drinking water, soil erosion, groundwater depletion, and salinization [52]. Substituting environmentally sound inputs for those which are damaging is an important step in addressing these problems [49]. In view of this, the Ministry of agriculture of China started the construction of the Key Laboratory of Agricultural Environment in the Huang-Huai-Hai Plain, with the objectives of scientific research, environmental monitoring, detection analysis and technical services, in 2012, aiming to carry out research on regional agricultural pollution prevention by means of agricultural non-point source pollution prevention, the environmental protection of producing areas, and the development and application of environment-friendly inputs. However, for the farmers who are the main body of agricultural production, whether these agricultural technology inputs will increase agricultural production costs and reduce agricultural income will be an important factor affecting the promotion of agricultural technology and the development of sustainable agriculture. Therefore, whether the economic, social and environmental benefits generated by agricultural production in the Huang-Huai-Hai Plain under the influence of agriculture technology can achieve a balance will be the focus of our future research.

## 6. Conclusions

In this paper, exploratory spatial data analysis, the gravity center model, and the spatial lag model were used to explore the spatial–temporal variation and influencing factors of the grain production pattern in the HHH from 1995 to 2018. The main conclusions were drawn as follows:

The grain production pattern in the HHH has the characteristics of being non-equilibrated in space and non-stationary in time. The spatial non-equilibrium is reflected in the shift of the grain production center from the southeast to the northwest of Tai’an city. The high-level areas of grain production capacity were mainly distributed in the east and south, while the low-level areas were distributed in the west and north. The non-stationarity of time is reflected in the rising trend in the grain production capacity and the weakening of the non-stationarity of time in the grain production center over time;

The global and local spatial agglomeration characteristics of grain production in the HHH were significant. The global spatial correlation characteristics underwent a “decrease–growth–decrease” change process, and the local spatial correlation characteristics demonstrated a concentrated distribution. Specifically, the hot spots were mainly distributed in the central and eastern regions of the HHH, and the cold spots were distributed in the north and southwest. The global and local spatial autocorrelation characteristics showed that the polarization of grain production in local areas has gradually weakened and the spatial difference has gradually decreased in the HHH, which indicates that its agricultural production has gradually shifted in the direction of sustainable development;

The impact of social–economic factors on grain production was constantly strengthened and the influence of climate factors on grain production was gradually weakened. EIA, AFA, and PCI helped to increase the grain yield per unit of sown area in the HHH; however, the effect of the PRE on grain production became weaker as time went on. We adopted the GWR model to prove that the EIA, AFA, and PCI had clear spatial heterogeneity in the intensity and direction of the local scale. The results showed that the EIA had a larger impact on grain production in the HHH compared with other factors, with the percentage of significance at 75.47%.

## Figures and Tables

**Figure 1 ijerph-17-09193-f001:**
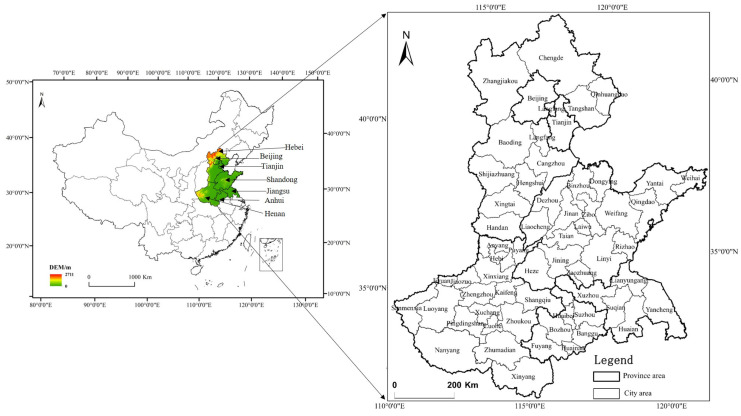
Location map of the Huang-Huai-Hai Plain (HHH).

**Figure 2 ijerph-17-09193-f002:**
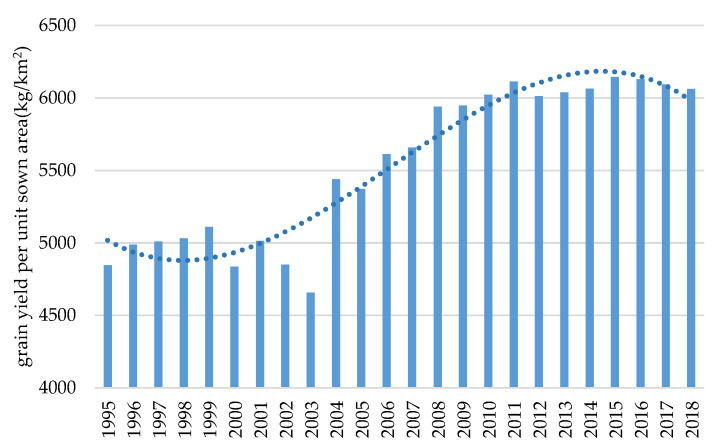
The temporal changes of grain production in the HHH from 1995 to 2018.

**Figure 3 ijerph-17-09193-f003:**
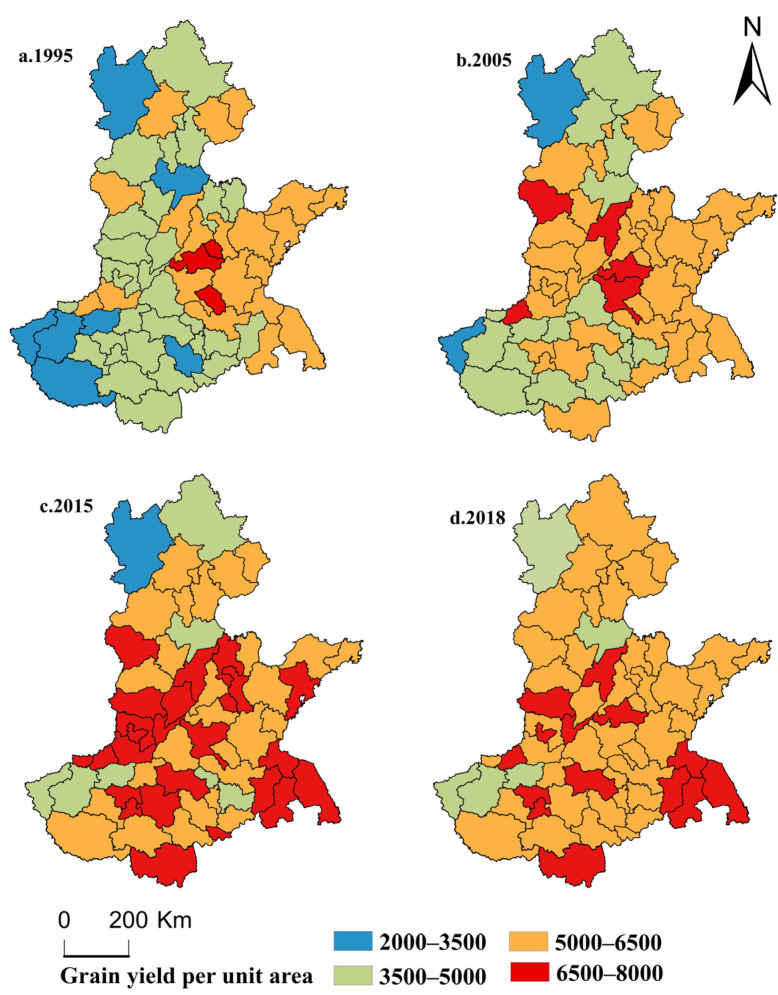
The spatial distribution of grain production in the HHH in (**a**) 1995, (**b**) 2005, (**c**) 2015, and (**d**) 2018.

**Figure 4 ijerph-17-09193-f004:**
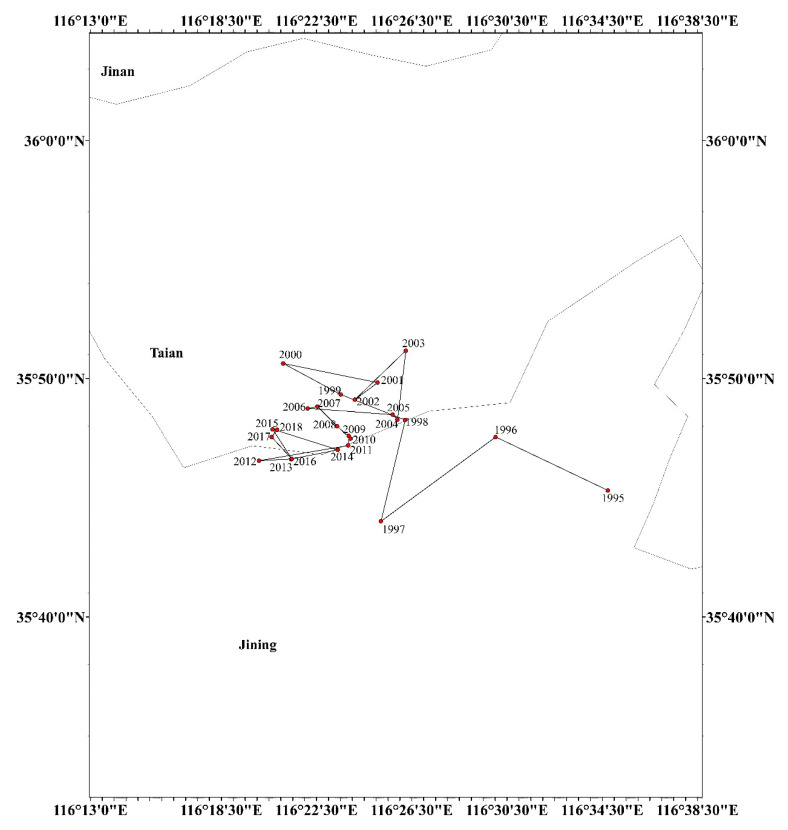
The track of the gravity center of grain production in the HHH from 1995 to 2018.

**Figure 5 ijerph-17-09193-f005:**
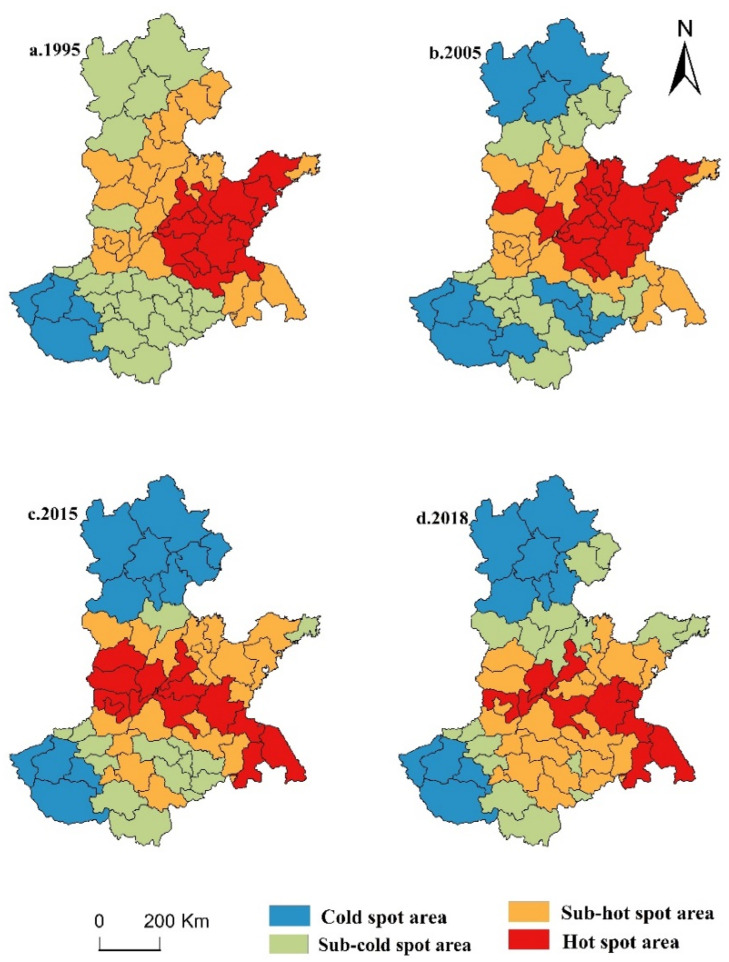
Spatial correlation characteristics of grain production in the HHH. (**a**) 1995, (**b**) 2005, (**c**) 2015, and (**d**) 2018.

**Figure 6 ijerph-17-09193-f006:**
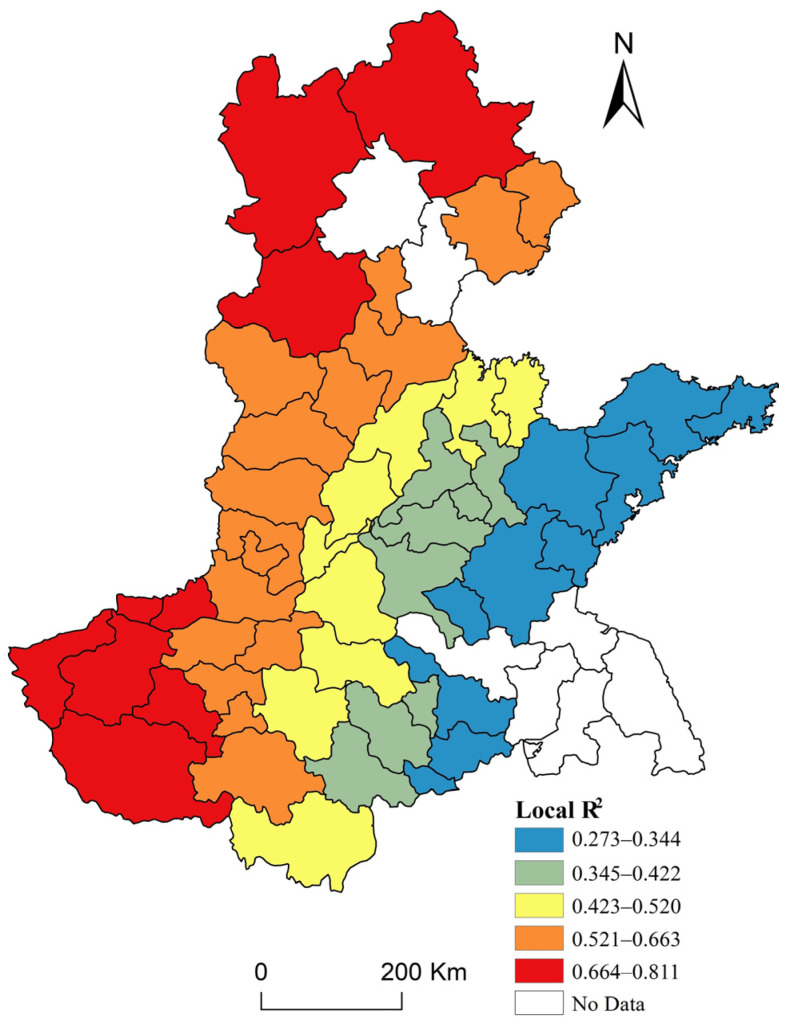
R-squared values derived from the geographical weighted regression (GWR) model.

**Figure 7 ijerph-17-09193-f007:**
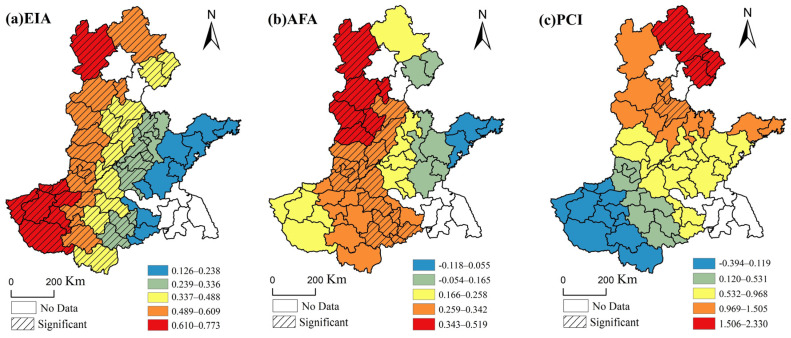
The spatial distribution of regression coefficients for the three factors based on a GWR model for 2018 in the HHH.

**Table 1 ijerph-17-09193-t001:** Index selection and specific treatment methods.

Data Type	Data Name	Processing Method	
Climate Data	Annual average temperature (TEM)	Kriging interpolation	°C
	Annual average precipitation (PRE)	Kriging interpolation	mm
	Annual average sunshine duration (SSD)	Kriging interpolation	h
Socio-economic Data	Proportion of effective irrigation area (EIA)	Effective irrigation area/arable land area	%
	Amount of fertilizer application per unit sown area (AFA)	Total fertilizer application/crop sown area	t/km^2^
	Amount of pesticide application per unit sown area (APA)	Total pesticide application/crop sown area	t/km^2^
	Mechanical power per sown area (MPA)	Agricultural machinery power/crop sown area	kw/km^2^
	Per capita annual income of rural residents (PCI)	/	RMB/per

**Table 2 ijerph-17-09193-t002:** The gravity center changes in grain production in 1995–2018.

Year	Gravity Center of Grain Production			Year	Gravity Center of Grain Production		
	Longitude	Latitude	Moving Distance/km		Longitude	Latitude	Moving Distance/km
1995	116.58 E	35.75 N	-	2007	116.38 E	35.81 N	0.774593
1996	116.50 E	35.79 N	9.663716	2008	116.39 E	35.80 N	2.163265
1997	116.42 E	35.73 N	11.05526	2009	116.40 E	35.79 N	1.415176
1998	116.44 E	35.80 N	8.099253	2010	116.40 E	35.79 N	0.261293
1999	116.39 E	35.82 N	5.382058	2011	116.40 E	35.79 N	0.747488
2000	116.35 E	35.84 N	5.081014	2012	116.33 E	35.78 N	7.035403
2001	116.42 E	35.83 N	7.488652	2013	116.36 E	35.78 N	2.505761
2002	116.41 E	35.82 N	2.213499	2014	116.39 E	35.78 N	3.691162
2003	116.44 E	35.85 N	5.510654	2015	116.34 E	35.80 N	5.316664
2004	116.43 E	35.80 N	5.391599	2016	116.36 E	35.78 N	2.751065
2005	116.43 E	35.80 N	0.532014	2017	116.34 E	35.79 N	2.319260
2006	116.37 E	35.81 N	6.620709	2018	116.35 E	35.80 N	0.697832

“-” means the item does not exist.

**Table 3 ijerph-17-09193-t003:** Moran’s *I* value of grain production in HHH.

Year	Moran’s *I*	*Z*-Score	*p*	Year	Moran’s *I*	*Z*-Score	*p*
1995	0.4114	5.0799	0.001 ***	2007	0.2371	3.0493	0.01 **
1996	0.3186	3.9572	0.001 ***	2008	0.2416	3.0646	0.01 **
1997	0.1718	2.2349	0.01 **	2009	0.2751	3.5401	0.001 ***
1998	0.3356	4.1347	0.001 ***	2010	0.2852	3.5766	0.001 ***
1999	0.3257	4.0797	0.001 ***	2011	0.2897	3.6144	0.001 ***
2000	0.3429	4.2799	0.001 ***	2012	0.2401	3.0216	0.01 **
2001	0.3286	4.1561	0.001 ***	2013	0.2529	3.1792	0.01 **
2002	0.2511	3.2294	0.001 ***	2014	0.3727	4.5792	0.001 ***
2003	0.5243	6.2996	0.001 ***	2015	0.3498	4.3184	0.001 ***
2004	0.2998	3.7368	0.001 ***	2016	0.2833	3.7924	0.001 ***
2005	0.3008	3.7651	0.001 ***	2017	0.2602	3.5101	0.001 ***
2006	0.2397	3.0403	0.01 **	2018	0.2315	3.1292	0.01 **

*** means significant at the 1% threshold level; ** means significant at the 5% threshold level.

**Table 4 ijerph-17-09193-t004:** Regression analysis results of the spatial lag model.

Model Variables	1995		2005		2015		2018	
	Coefficient	Standard Deviation	Coefficient	Standard Deviation	Coefficient	Standard Deviation	Coefficient	Standard Deviation
Constant	0.036	1.433	5.516	0.914	5.215 ***	1.155	8.665 ***	1.749
EIA	0.505 ***	0.072	0.415 ***	0.048	0. 532 ***	0.081	0.588 ***	0.099
AFA	0.008 **	0.004	0.208 ***	0.053	0.124 **	0.053	0.023 **	0.029
PCI	0.405 ***	0.109	0.170 **	0.068	0.124 *	0.090	0.050 *	0.069
PRE	0.508 ***	0.170	0.016	0.794	0.148	0.099	0.022	0.061
R^2^	0.729		0.779		0.656		0.529	
AIC	−34.039		−87.236		−85.007		−95.487	

*** means significant at the 1% threshold level; ** means significant at the 5% threshold level; * means significant at the 10% threshold level.

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
