# Peer review of "Spatial-Temporal Characteristics in Grain Production and Its Influencing Factors in the Huang-Huai-Hai Plain from 1995 to 2018"

_ijerph, 2020, doi:10.3390/ijerph17249193_

Round 1
Reviewer 1 Report
Food is critical for China. Grain production is a fundamental issue and a critical point for China’s food security. As far as I know, the Huang-Huai-Hai (HHH) plain is one of the major grains producing areas and sensitive to climate change and socio-economic activities. The paper addressed the spatiotemporal pattern of the grain production in the HHH plain and try to attribute its association with climate factors and socio-economic factors. Although the research object is critical, and some results were obtained, I still have a few concerns that need to be cleared.
- The introduction requires an extensive revision. Since this article focused on the grain production of the HHH plain, more detailed background information, relevant literature, and recent advances regarding the HHH grain production should be thoroughly summaries.
- The authors should provide clearer figures and equations. They are very blurring and can not be read and understand under existing quality. Readable and understanding figures are the basic condition for eligible publications.
- According to the authors, the spatiotemporal pattern of the grain production in the HHH plain actually relied on the classification of the grain production per unit area sown. What is the basis for the authors to divide the grain yield into 2000-3500, 3500-5000,5000-6500, and 6500-8000 kg/km2? What is the statistical distribution of this data in the HHH from 1995,2005,2015, and 2018? If these classification criteria changed, what are the spatiotemporal pattern?
One more issue is the time selection. From 1995 to 2018, grain production many have distinct evolution. Why the authors selected these four times? What is the whole temporal development of the grain productions in the HHH region from 1995 to 2018?
- Although the authors make a lot of effort to explain the socio-economic factor influencing grain production, the climate factors are not convincible. Besides the precipitation, the annual temperature and annual sunshine time also significantly affect grain production. The authors described temperature and sunshine duration data. However, the results and discussions lack these two factors.
Other detailed comments listed as follows:
- Line 21-22. What is the development trend of local spatial agglomeration?
- Line 30. What does the “local geographical level” represent?
- Line 36. “China’s food self-sufficiency rate has reached more than 95%” need a source or reference
- Line 40, Where does the data “1.53℃” come from?
- Line 49-53. The description of the sentence is not clear.
- Line 51. The references’ format is not right. Please thoroughly check out.
- Line 142. The data’s link is not working.
- Line 183. The abbreviation SLM should be the Spatial Lag Model.
- Line 196, Please give the full name of AICc
- Line 216-217. This sentence should be moved to the next paragraph.
- Line 219- 227. The description of this paragraph needs improvement.
- Line 271. Typo. “Tai an” should be Tai’an.
- Line 276. “P test value” should be P-value.
- Line 344. The citation’s format of the Figure and Table should be consistent.
Author Response
Point 1: The introduction requires an extensive revision. Since this article focused on the grain production of the HHH plain, more detailed background information, relevant literature, and recent advances regarding the HHH grain production should be thoroughly summaries.
Response 1: Thank you very much for your suggestion and we agree with you. The background information, relevant literature, and recent advances of the HHH grain production are important for the research. So we added the background, relevant literature, and recent advances of the HHH grain production, and cut some contents that are not relevant enough. Therefore, the introduction part is adjusted as follows:
1.Introduction
China is an important food producing country in the world, as well as a large food consumer. China’s food self-sufficiency rate has reached more than 95%[1]. Although the current grain supply and demand in China maintains a balance of total quantity and a surplus in harvest, the small per capita arable land area, low mechanization level, and family-based business units determine that the current arable land has an extremely limited potential for increasing grain production [2, 3].From 1850–1900 to 2006–2015, the mean land surface air temperature increased by 1.53 ℃[4]. Climate change has already affected food security due to warming, changing precipitation patterns, and the greater frequency of some extreme events [4]. Under the dual constraints of climate change and the process of urbanization, unpredictable meteorological disasters, limited arable land resources, huge population pressure, and the diversified consumer demand of residents directly generated strong demand for stable grain production [5, 6]. Additionally, the outbreak of COVID-19 in 2019 led to labor shortages and supply chain disruptions, which affected the food security of some countries and regions. Certain countries have even banned the export of food, leading to large fluctuations in global food prices and casting a shadow over the world’s food crisis. Although China’s major food crops, such as rice and wheat, are less dependent on the international market and the national food security will not be affected by the agricultural trade restrictions brought about by the COVID-19, ensuring the China’s food security in the later period of the epidemic is of great strategic significance for ensuring domestic economic recovery and social stability.
The Huang-Huai-Hai Plain (HHH), located in the north of the country, is a high-yield agricultural region[8], accounting for 70% of wheat and 30% of maize production in China[9]. Because of its importance in grain production and China’s self-sufficiency in food, it is known as the ‘Breadbasket of China’[10, 11]. Changes in grain production in the HHH can have direct impacts on both the national economy and food security of China[12]. Therefore, it is of great significance to understand the temporal and spatial characteristics and its influencing factors of grain production in the Huang Huai Hai plain to ensure national food security.
Previous studies have pointed out that while grain production in the Huang Huai Hai Plain is increasing, the difference in grain yields in various regions is gradually shrinking, but the spatial distribution pattern of grain production in the southern region of the Huang Huai Hai Plain is higher than that in the northern region remains unchanged [13-15]. Later studies attribute the distribution characteristics of grain production in the HHH to socio-economic factors[13, 14, 16]. The increase in grain production in the HHH is mainly due to the improvement of crop varieties, fertilizers and effective irrigation area. For example, the amount of fertilizer used has increased by about 400% and the effective irrigation area has increased by about 20% [10]. Liu, Tang [15] used the spatial lag model to reveal the factors affecting the differentiation of grain yield in the HHH from 1995 to 2010, and found that farmers' per capita net income, effective irrigation area ratio, and industrial structure had significant positive effects on grain yield. At the same time, studies have also found that although grain production and fertilizer input in the HHH are still significantly positively correlated, the current application of fertilizers in actual production is extremely unbalanced and unreasonable [17]. It is essentially important to improve fertilizer use efficiency for save resources as well as increase yield [10].
A number of studies have also emphasized the role of climate factors, especially precipitation, on grain production in the HHH [18-22]. Qu, Li [20] indicated that the increase in precipitation in the HHH can significantly increase the food production in the HHH, but the increase in thermal resources will increase the shortage of water resources and offset the impact of the increase in temperature. Xiao, Qi [23] reveled that climate change reduced the potential winter wheat yield of 80% of the stations by 2.3-58.8 kg∙yr-1,at the same time, it is pointed out that increasing the heat time of wheat growth period is essential to alleviate the impact of the shortening of growth period caused by warming climatic conditions. But if the advancement of agricultural technology and other non-climatic factors are taken into account, for every 1°C increase in the average temperature of the Huang Huai Hai Plain, the winter wheat yield in the north will increase by 2.1%, and the yield in the south will decrease by 4.0% [22].
Therefore, most of previous research on grain production in the Huang Huai Hai Plain mainly focuses on the factors affecting grain production, that is, mainly from two aspects: socio-economic factors and climatic factors. And these studies are only carried out from one level, which separates the comprehensive impact of climate and socio-economic factors on grain production. This will inevitably affect the final assessment results. To fill these research, this paper aims to explore the influencing factors of grain production in HHH by incorporating the climate factors and socio-economic factors into the models to provide a reference for ensuring food security and relevant departments to make decisions.
Point 2: The authors should provide clearer figures and equations. They are very blurring and can not be read and understand under existing quality. Readable and understanding figures are the basic condition for eligible publications.
Response 2: Thank you very much for your good suggestion and we agree with you. I’m very sorry these errors have affected your reading feelings. We have re-provided clear figures and use the Equation Editor to edit the formula equations. Details are as follows:
3.1. Exploratory Spatial Data Analysis (ESDA)
Line 219
|
(1) |
Line 229
|
(2) |
Line 239
|
(3) |
Line 244
|
(4) |
3.2. Gravity Center Model
Line 267
|
|
(5) |
Line 274
|
(6) |
3.3. Spatial Panel Data Model
Line 284
|
(7) |
|
|
(8) |
3.4. Geographically Weighted Regression (GWR) Model
Line 293
|
(9) |
4.12. Spatial characteristics of grain production in the HHH
Line 352
Figure 3. The spatial distribution of grain production in the HHH in 1995, 2005, 2015, and 2018.
4.3. Dynamic Change of gravity center of Grain Production in the HHH
Line 382
Figure 4. The track of the gravity center of grain production in the HHH from 1995 to 2018.
4.3.2. Local spatial correlation characteristics
Line 454
Figure 5. Spatial correlation characteristics of grain production in the HHH.
4.4.2. Analysis of spatial heterogeneity of influencing factors
Line 519
Figure 6. R-squared values derived from the geographical weighted regression (GWR) model.
Line 553
Figure 7. The spatial distribution of regression coefficients for the three factors based on a GWR model for 2018 in the HHH.
Point3: According to the authors, the spatiotemporal pattern of the grain production in the HHH plain actually relied on the classification of the grain production per unit area sown. What is the basis for the authors to divide the grain yield into 2000-3500, 3500-5000,5000-6500, and 6500-8000 kg/km2? What is the statistical distribution of this data in the HHH from 1995,2005,2015, and 2018? If these classification criteria changed, what are the spatiotemporal pattern?
Response 3: Thank you for your suggestion and we agree with you. We chose this classification for the following reasons:
Firstly, the classification results are obtained by using the equal interval classification method. Equal interval classification is most suitable for common data ranges. This method emphasizes the amount of a certain attribute value relative to other values. Compared with other classification methods, this method does not place similar features in adjacent classes, nor does it place features with larger values in the same class, so the resulting distribution map will not be misleading. Secondly, the grain yield in the study area ranges from 2000 to 8000, and we use the equal interval classification method to divide it into 4 categories, namely 2000-3500, 3500-5000, 5000-6500, and 6500-8000 kg/km2. This is also convenient for comparison at four points in time. Finally, the classification method is just a tool, even if the classification standard is changed, the spatial distribution pattern of grain production in the HHH Plain will not change. The areas with high grain yield per unit sown area of the Huang-Huia-Hai Plain were still mainly distributed in the east and south, while the areas with low grain yield per unit sown area were still mainly distributed in the west and north of the HHH.
Point4: One more issue is the time selection. From 1995 to 2018, grain production many have distinct evolution. Why the authors selected these four times? What is the whole temporal development of the grain productions in the HHH region from 1995 to 2018?.
Response 4: Thank you for your suggestion and we agree with you. We add the whole temporal development of the grain productions in the HHH region from 1995 to 2018 in 4.1. Details are as follows:
Line 302-312:
4.1. Temporal changes of grain production in the HHH
From 1995 to 2018, the grain yield per unit sown area in the HHH has shown a steady increase (Fig. 2), which can be divided into three stages: fluctuating growth stage (1995-2005), steady growth stage (2005-2015), slow descent stage (2015-2018). Firstly, in the fluctuating growth stage (1995-2005), although the grain yield per unit sown area has increased in this stage, the fluctuation range is relatively large and the grain yield per unit sown area gradually stabilized in 2005. Secondly, during the steady growth stage (2005-2015), the grain yield per unit sown area showed a characteristic of small fluctuation growth. Finally, a slow decline began to appear in the grain yield per unit sown area in the HHH during the slow descent stage (2015-2018).
Figure 2. The temporal changes of grain production in the HHH from 1995 to 2018
Therefore, according to the three different stages of development of grain yield in the HHH plain, we chose corresponding turning points in time, namely 1995,2005,2015,2018.
Point5: Although the authors make a lot of effort to explain the socio-economic factor influencing grain production, the climate factors are not convincible. Besides the precipitation, the annual temperature and annual sunshine time also significantly affect grain production. The authors described temperature and sunshine duration data. However, the results and discussions lack these two factors.
Response 5: Thank you for your suggestion and we agree with you. The annual temperature and annual sunshine time also significantly affect grain production. So we add the two factors in the results and discussions. Details are as follows:
Line 587-596
However, at the same time, we should also be aware of the relationship between rainfall and groundwater; that is, rainfall becomes groundwater through infiltration, providing sufficient water for irrigating crops. With the improvement of irrigation facilities, irrigation has gradually become an indispensable and important means for stable grain production. This may also be the reason why the direct impact of PRE on grain production is gradually weakening and why the EIA is gradually increasing. Future research should focus on related research in this area. The reasons for the insignificance effect of temperature and sunshine duration may be attributed to two points: first, compared with precipitation, the heat resources of the HHH Plain can well meet the growth needs of winter wheat and summer corn, and the yield is less affected by temperature and sunshine duration. At the same time, an increase in temperature and sunshine duration may increase evaporation, thereby offsetting the effect of precipitation. It is also possible to attribute this effect to precipitation rather than temperature and sunshine duration. Second, it may be that the crops in the research area are not subdivided. Different crops have inconsistent requirements for temperature and sunshine duration, which may weaken the effects of temperature and sunshine duration. This is also the focus of future research.
Other detailed comments:
Point 1: Line 21-22. What is the development trend of local spatial agglomeration?
Response 1: Thank you very much for your good suggestion and we agree with you. We complete the description of the development trend of local spatial agglomeration. Details are as follows:
Line 20-24
(3) The global spatial autocorrelation experienced the changing process of "decline–growth–decline" and the area of hot and cold spots is gradually reduced and stabilized,
which indicated that the polarization of grain production in local areas has gradually weakened and the spatial difference has gradually decreased in the Huang-Huai-Hai Plain.
Point 2: Line 30. What does the “local geographical level” represent?
Response 2: Thank you very much for your good suggestion. R-squared represents the local geographical level, which means that these three variables have passed the geographical variability test, that is, there is significant spatial heterogeneity at the local geographic level.
Point 3: Line 36. “China’s food self-sufficiency rate has reached more than 95%” need a source or reference.
Response 3: Thank you very much for your good suggestion and we agree with you. We add the reference about this sentence. Details are as follows:
Line 39
China’s food self-sufficiency rate has reached more than 95%[1]
Point 4: Line 40, Where does the data “1.53℃” come from?
Response 4: Thank you very much for your good suggestion and we agree with you. We add the reference about this sentence. Details are as follows:
Line 43
From 1850–1900 to 2006–2015, the mean land surface air temperature increased by 1.53 ℃[4]
Point 5: Line 49-53. The description of the sentence is not clear.
Response 5: Thank you very much for your good suggestion and we agree with you. We re-described this sentence. Details are as follows:
Line 54-56
Although China’s major food crops, such as rice and wheat, are less dependent on the international market and the national food security will not be affected by the agricultural trade restrictions brought about by the COVID-19, ensuring the China’s food security in the later period of the epidemic is of great strategic significance for domestic economic recovery and social stability.
Point 6: Line 51. The references’ format is not right. Please thoroughly check out.
Response 6: Thank you very much for your good suggestion and we agree with you. We have carefully checked the format of the references as required by the journal.
Point 7: Line 142. The data’s link is not working.
Response 7: Thank you very much for your good suggestion and we agree with you. We re-provided the data’s link in line 191. Details are as follows:
Line 190
The historical climate data were collected from the Chinese meteorological data hub (https://data.cma.cn).
Point 8: Line 183. The abbreviation SLM should be the Spatial Lag Model.
Response 8: Thank you very much for your good suggestion and we agree with you. We corrected the abbreviation SLM. Details are as follows:
Line 283
Instead, the spatial lag model (SLM) or the spatial error model (SEM) should be used for analysis.
Point 9: Line 196, Please give the full name of AICc.
Response 9: Thank you very much for your good suggestion and we agree with you. We have gave the full name of AICc. Details are as follows:
Line 299
The optimal bandwidth distance can be obtained automatically in GWR4.0 corrected by finite correction of the Akaike Information Criterion (AICc).
Point 10: Line 216-217. This sentence should be moved to the next paragraph.
Response 10: Thank you very much for your good suggestion and we agree with you. We have moved that sentence to the next paragraph. Details are as follows:
Line 333-334
The unit area sown to grain production maintained in the 6500–8000 kg/km2 interval number of urban change trend was different from the above three kinds. In 1995, the number of cities within this range was 3, which increased to 5 in 2005, 24 in 2015, and reduced to 14 in 2018, which reflects a characteristic of a sharp increase followed by a slow decline. But they were still mainly distributed in the central and southern parts of the HHH.
Point 11: Line 219- 227. The description of this paragraph needs improvement.
Response 11: Thank you very much for your good suggestion and we agree with you. This paragraph has been improved. Details are as follows:
Line 336-337
In 1995, the number of cities within this range was 3, which increased to 5 in 2005, 24 in 2015, and reduced to 14 in 2018, which reflects a characteristic of a sharp increase followed by a slow decline. But they were still mainly distributed in the central and southern parts of the HHH.
Point 12: Line 271. Typo. “Tai an” should be Tai’an.
Response 12: Thank you very much for your good suggestion and we agree with you. We have corrected this word. Details are as follows:
Line 394
The center was still stable in the territory of Tai’an city, that is, to the southeast of the HHH.
Point 13: Line 276. “P test value” should be P-value.
Response 13: Thank you very much for your good suggestion and we agree with you. We have corrected the description. Details are as follows:
Line 392
Based on the grain yield data per unit sown area, the Moran's I value, Z statistic, and P-value were calculated using Geoda software, and the spatial correlation characteristics of grain production are shown in Table 3.
Point 14: Line 344. The citation’s format of the Figure and Table should be consistent.
Response 14: Thank you very much for your good suggestion and we agree with you. We have corrected the citation’s format of the Figure and Table. Details are as follows:
Line 467
According to Table 4, the coefficients of EIA were 0.505, 0.415, 0.532, and 0.588 in 1995, 2005, 2015, and 2018, and were effective at the 1% threshold level.

Reviewer 2 Report
The paper is within the scope of the journal's subject matter. It provides a spatial-temporal analysis of grain production and factors influencing it in the Huang-Huai-Hai plain in 1995-2018.
The abstract is formulated correctly and contains the necessary elements.
The article used exploratory spatial data analysis, the center of gravity model and the spatial lag model to study the spatial-temporal variability and factors influencing the grain production model in HHH from 1995 to 2018.
The authors point to the decreasing importance of climatic factors on the production of cereals and the increasing importance of socio-economic factors. In particular, they indicate a positive impact on the production volume of the following variables: the ratio of effectively irrigated area, the amount of fertilizer used per unit area of sown area and the average annual per capita income of rural residents. The analysis is carried out correctly and the conclusions in the scope offered by the authors are exhaustive.
However, for the analysis to be complete, it is necessary to take into account the following remarks:
1) The studied area is quite specific, and the dependencies discussed indicate the desire to intensify production and (perhaps) its industrialization rather. Therefore, it seems necessary to refer to this problem as well as to try to answer the question whether this is the right direction of development?
2) The results presented indicate the limited use of the production potential, and above all, almost complete lack of reference to the relation between production targets and achieving sustainable agricultural production in the Pareto sense. It is valuable for the Authors to address this issue and try to answer the question (or at least consider the problem): what is the relationship between ignoring the issue of sustainability in agricultural policy and both clear spatial heterogeneity of the production volume and its changes over time? Can sustainability be disregarded in the long run without sharp production slumps or environmental disasters? How does irrigation affect local ecosystems?
3) In order to take into account these problems, it is necessary to develop the literature review and discussion on issues related to sustainable agriculture.
Author Response
Point 1:
1) The studied area is quite specific, and the dependencies discussed indicate the desire to intensify production and (perhaps) its industrialization rather. Therefore, it seems necessary to refer to this problem as well as to try to answer the question whether this is the right direction of development?
2) The results presented indicate the limited use of the production potential, and above all, almost complete lack of reference to the relation between production targets and achieving sustainable agricultural production in the Pareto sense. It is valuable for the Authors to address this issue and try to answer the question (or at least consider the problem): what is the relationship between ignoring the issue of sustainability in agricultural policy and both clear spatial heterogeneity of the production volume and its changes over time? Can sustainability be disregarded in the long run without sharp production slumps or environmental disasters? How does irrigation affect local ecosystems?
3) In order to take into account these problems, it is necessary to develop the literature review and discussion on issues related to sustainable agriculture.
Response 1: Thank you very much for your suggestion and we agree with you. Sustainable agriculture and agricultural modernization are the main considerations and development directions for the Huang Huai Hai Plain as a major food production base in the future. Therefore, we added the literature review and discussion on issues related to sustainable agriculture. Details are as follows:
Line 66-80
Previous studies have pointed out that while grain production in the Huang Huai Hai Plain is increasing, the difference in grain yields in various regions is gradually shrinking, but the spatial distribution pattern of grain production in the southern region of the Huang Huai Hai Plain is higher than that in the northern region remains unchanged [17-19]. Later studies attribute the distribution characteristics of grain production in the HHH to socio-economic factors[17, 18, 20]. The increase in grain production in the HHH is mainly due to the improvement of crop varieties, fertilizers and effective irrigation area. For example, the amount of fertilizer used has increased by about 400% and the effective irrigation area has increased by about 20% [10]. Liu, Tang [19] used the spatial lag model to reveal the factors affecting the differentiation of grain yield in the HHH from 1995 to 2010, and found that farmers' per capita net income, effective irrigation area ratio, and industrial structure had significant positive effects on grain yield. At the same time, studies have also found that although grain production and fertilizer input in the HHH are still significantly positively correlated, the current application of fertilizers in actual production is extremely unbalanced and unreasonable [21]. It is essentially important to improve fertilizer use efficiency for save resources as well as increase yield [10].
Line 609-640
However, the impact of relevant agricultural policies issued by the country also needs to be considered in the future. From 1995 to 2018, the overall increase in grain production in HHH and the gradual reduction of spatial differences well reflect the background and national policies of the country in different periods. Since 1995, with the advancement of agricultural technology, the agricultural development of the HHH has been weakened by natural conditions, and the grain production has increased significantly[17]. At the same time, due to the popularization of fertilizers, pesticides, and irrigation, the gap in grain production in various regions in HHH is also narrowing.
Due to rapid urbanization, the conversion of fertile irrigated land to non-agricultural land seems to pose a potential threat to the food security of the HHH - even the whole of China[10]. Moreover, farmers are gradually migrating to cities in search of higher incomes due to the urban-rural development gap[57], which has caused the arable land in the HHH to be abandoned, resulting in the expansion of spatial differences in grain production in different regions in the HHH, and also threatening national food security. In order to ensure national food security, the central government proposed the construction of "high-standard basic farmland projects" and "agricultural modernization" to promote the large-scale, intensive, and modernized management of arable land in the HHH, thereby increasing the grain production of the HHH, and promoting the development of sustainable agriculture.
At the same time, we should not ignore the resistance caused by resource endowments to the sustainable agriculture in the HHH. Especially, the precipitation cannot meet the water demand of the crops[8, 58], and water shortage is one of the major factors threatening high and stable production of wheat[13]. The water consumption greatly exceeds the precipitation and groundwater must be extracted to make up for the deficiency to maintain high yield[14]. Some areas in the HHH even appear salinization due to unreasonable irrigation[10], which poses huge challenges for the minimization of environmental impacts and development of sustainable agriculture[15, 59]. Therefore, the government must not only build irrigation facilities, but more importantly, promote water-saving irrigation technologies, improve water resource utilization efficiency, implement drip irrigation, sprinkler irrigation[15], etc., so as to achieve sustainable agricultural production in the HHH. At the same time, in the long run, in the face of the encroachment of arable land in the promotion of urbanization, the amount of arable land in the future will also face severe challenges[10]. Promoting intensive agricultural production and improving the level of intensive use of agricultural production are also of great significance to ensure future food production[60]. Moreover, it can also enhance the ability to respond to natural disasters in the future.

Reviewer 3 Report
Page 5, Methods Section: The authors provide a general description of the statistics calculated for their exploratory spatial data analysis. However, this section is a bit brief. Readers unfamiliar with these statistics will likely have a hard time following the authors interpretation of these statistics as reported later in the manuscript. The equations are straight forward, but the authors need to provide more explanation about each statistic and how each statistic is used. For example, what is a “geometric barycenter point” and how is this statistic used? What are the global and local spatial correlation degrees that are estimated by the Moran’s I and Getis-ORD Gi statistics, respectively. A better explanation of these statistics are and how they are used would help the reader understand the results presented later in the manuscript. This section should be expanded.
Page 6, Line 185: A slight typo. Instead of referring to y as “the explain variable”, the authors should write “the dependent variable.” Also, x on line 185 is a matrix of explanatory variables, so instead of saying “x is the explanatory variable,” the authors should say “x are the explanatory variables.”
Page 8, Figure 3, Table 2 and Pages 8 – 9 Discussion, Lines 264 - 298: Again, it’s hard for the reader to know what the information in Figure 3 and Table 2 means without knowing what a barycenter is. Similarly, it is hard for the reader to follow the results of the Moran’s I statistic without fully understanding what this statistic means.
Pages 9 - 10, Lines 301 – 330 and Page 10, Figure 4: The authors comment on and present in Figure 4 “hot spots,” “cold spots,” “sub-hot spots,” and “sub-cold spots. It would be good for the authors to explain what “hot spots” and “cold spots” are.
Page 12, Lines 383 - 384: The authors state the following: “The three factors—EIA, AFA, and PCI—were put into the model according to the criteria of AICc minimization.” What do the authors mean by “AICc minimization”?
Page 14, Line 450: The authors write “The results showed that the EIA had a larger impact on explain grain production in the HHH compared with other factors, with the percentage of significance at 75.47%.” The word “explain” appears to be a typo in the sentence. Perhaps the authors should state the following: “… the EIA had a larger impact on the grain production dependent variable in the HHH…”, or maybe they should simply delete “explain” from the sentence.
Author Response
Point 1: Page 5, Methods Section: The authors provide a general description of the statistics calculated for their exploratory spatial data analysis. However, this section is a bit brief. Readers unfamiliar with these statistics will likely have a hard time following the authors interpretation of these statistics as reported later in the manuscript. The equations are straight forward, but the authors need to provide more explanation about each statistic and how each statistic is used. For example, what is a “geometric barycenter point” and how is this statistic used? What are the global and local spatial correlation degrees that are estimated by the Moran’s I and Getis-ORD Gi statistics, respectively. A better explanation of these statistics are and how they are used would help the reader understand the results presented later in the manuscript. This section should be expanded.
Response 1: Thank you very much for your suggestion and we agree with you. We provided a more detailed introduction to the method. Details are as follows:
- Methods
3.1. Exploratory Spatial Data Analysis (ESDA)
Exploratory spatial data analysis is a collection of techniques for describing and visualizing spatial distributions; determining atypical locations or spatial outliers; discovering spatial associations, clusters, or hot spots; it is also used to imply spatial characteristics or other forms of space heterogeneity[43]. In general, global and local spatial autocorrelation (or hot spots analysis) are often used to explore the spatial characteristics of observations[44].
3.1.1. Global spatial autocorrelation
Global spatial autocorrelation is used to test the spatial correlation of the observations of spatial units within the study area[45], and is mainly measured by the Global Moran's I, which was first proposed by Moran[46]. The Moran's I can be calculated using Eq.(1):
(1)
where I represents Moran’s I; n is the number of spatial units (in this study, n=59); xi and xj are the observations of spatial units I and j, respectively; is the average value of observations of spatial units; wij is the spatial weight matrix, where wij =1 if spatial units I and j share a common border and wij= 0 otherwise. Values of Global Moran's I range from−1 to 1. If I<0, it means there is a negative spatial correlation in the space; if I>0, it means there is a positive spatial correlation; if I=0, it means there is no spatial correlation.
The significance of Moran's I is usually measured by Z statistics using Eq.(2):
(2)
where E(I) and Var(I) are the expected value and variance of Moran's I, respectively.
3.1.2. Hot spot (Getis-Ord Gi*) analysis
The Getis-Ord Gi* is commonly used for hotspot analysis, which can identify clustering relationships at different spatial locations. Compared with the local spatial autocorrelation, the Getis-Ord Gi* is more sensitive to the identification of cold and hot spots, and can fully reflect the high or low value distribution relationship between a certain geographic element and other surrounding elements[47]. The formula is [47-50]:
(i≠j) (3)
where , n is the number of spatial units (in this study, n=59); wij is the spatial weight matrix, where wij =1 if spatial units I and j share a common border and wij= 0 otherwise.
The significance of Gi* is usually measured by Z statistics using Eq.(4):
(4)
where E(Gi*) and Var(Gi*) are the expected value and variance of Gi*, respectively. If Z(Gi*) is significantly positive, it indicates that the observations around the spatial unit i are relatively high (higher than the average), and are high-value clusters in the space, belonging to hot spots; on the contrary, if Z(Gi*) is significantly negative, it indicates that the observations around the spatial unit i are relatively low (lower than the mean), and are low-value clusters in the space, belonging to cold spots. The larger (or smaller) the Z(Gi*) is, the more intense the clustering of high (or low) values. A Z(Gi*) near zero indicates no apparent spatial clustering.
3.2. Gravity Center Model
The gravity center model is used to measure the overall distribution of a certain attribute in a region. It can provide a concise and accurate feature of the distribution of the attribute in the space, and can indicate the general trend and central location of its distribution. We assumed that a large region (such as an administrative region) consists of several subregions, and then the gravity center of grain production in the region can be calculated by the grain production and geographic coordinates of each sub-region. The formula is[51]:
(2)
In equation (2), Xi,Yi represents the geographic coordinates of the ith subregion. Mi represents the grain yield per unit sown area of the subregion. X and Y represent the gravity center of grain production in a large region. Using formula (3), the moving distance of the gravity center in grain production can be obtained, which can reflect the evolution of the gravity center of a property in a region
(i>j) (3)
In equation (3), Dij is the gravity center movement distance (km) of grain production from j to i years. (Xi, Yi) and (Xj, Yj) are the gravity center coordinates of grain production in the i and j years. R is typically 111.111, which represents the coefficient of spherical longitude and latitude coordinates converted to plane distance.
Point 2: Page 6, Line 185: A slight typo. Instead of referring to y as “the explain variable”, the authors should write “the dependent variable.” Also, x on line 185 is a matrix of explanatory variables, so instead of saying “x is the explanatory variable,” the authors should say “x are the explanatory variables.”.
Response 2: Thank you very much for your good suggestion and we agree with you. I’m very sorry these errors have affected your reading feelings. We have corrected this error. Details are as follows:
Line 286
In Equations (4) and (5), y is the dependent explain variable, x are the explanatory variables, Wij is the space weight matrix, ρ is the spatial hysteresis parameter, β is the parameter vector, μ is the random interference term, ε is the regression residual vector, and λ is the autoregression parameter.
Point3: Page 8, Figure 3, Table 2 and Pages 8 – 9 Discussion, Lines 264 - 298: Again, it’s hard for the reader to know what the information in Figure 3 and Table 2 means without knowing what a barycenter is. Similarly, it is hard for the reader to follow the results of the Moran’s I statistic without fully understanding what this statistic means.
Response 3: Thank you for your suggestion and we agree with you. What makes it hard for the reader to understand is a bit brief of the method description. So we provide a better explanation of these statistics are and how they are used, which would help the readers understand the results presented in the manuscript. Details are the same as the point 1.
Point 4: Pages 9 - 10, Lines 301 – 330 and Page 10, Figure 4: The authors comment on and present in Figure 4 “hot spots,” “cold spots,” “sub-hot spots,” and “sub-cold spots. It would be good for the authors to explain what “hot spots” and “cold spots” are.
Response 4: Thank you for your suggestion and we agree with you. We add the explanation of hot spots” and “cold spots”. Details are as follows:
Line 246-253:
where E(Gi*) and Var(Gi*) are the expected value and variance of Gi*, respectively. If Z(Gi*) is significantly positive, it indicates that the observations around the spatial unit i are relatively high (higher than the average), and are high-value clusters in the space, belonging to hot spots; on the contrary, if Z(Gi*) is significantly negative, it indicates that the observations around the spatial unit i are relatively low (lower than the mean), and are low-value clusters in the space, belonging to cold spots.
Point5: Page 12, Lines 383 - 384: The authors state the following: “The three factors—EIA, AFA, and PCI—were put into the model according to the criteria of AICc minimization.” What do the authors mean by “AICc minimization”?
Response 5: Thank you for your suggestion and we agree with you. AICc is mainly used to determine the optimal bandwidth of the GWR model. The smaller the AICc value, the better the fit of the GWR model. Therefore, the minimum value of AICc is selected, that is, the model with the best fitting effect is selected for analysis. We also add a sentence to explain the AICc in 301-302. Details are as follows:
Line 299-300
The smaller the AICc value, the higher the goodness of fit of the model will be[56].
Point 6: Page 14, Line 450: The authors write “The results showed that the EIA had a larger impact on explain grain production in the HHH compared with other factors, with the percentage of significance at 75.47%.” The word “explain” appears to be a typo in the sentence. Perhaps the authors should state the following: “… the EIA had a larger impact on the grain production dependent variable in the HHH…”, or maybe they should simply delete “explain” from the sentence.
Response 6: Thank you very much for your good suggestion and we agree with you. We delete “explain” from the sentence. Details are as follows:
Line 580
The results showed that the EIA had a larger impact on explain grain production in the HHH compared with other factors, with the percentage of significance at 75.47%.

Round 2
Reviewer 1 Report
Clear and beautiful figures are a key factor and a basic requirement for a qualified research article. The revised figures are still too blurring to read! Please make a real improvement!
Author Response
Point 1: Clear and beautiful figures are a key factor and a basic requirement for a qualified research article. The revised figures are still too blurring to read! Please make a real improvement.
Response 1: Thank you very much for your good suggestion and we agree with you. I’m very sorry these errors have affected your reading feelings. Firstly, we set the image resolution of the pictures in the article to 600 dpi to ensure the clarity of the pictures. Secondly, in view of the number of cities, it will not be clear if the city name is displayed on the map. Therefore, we adjusted the location map to enlarge the image size of the Huang-Huai-Hai Plain, and marked the city names on the location map. Then, we deleted the city names on the spatial distribution maps in the paper to ensure the clarity of location and boundary of the cities. Finally, because the clarity of the pictures inserted into the document may be slightly affected, we also provided the original pictures to the journal editor. Details are as follows:
Line 181
Line 353
Line 384
Line 456
Line 521
Line 554

Reviewer 2 Report
The amendments done improved the article, but the authors have not yet discussed the issue of sustainability in three parallel dimensions: economic, social and environmental. The authors state that "Promoting intensive agricultural production and improving the level of intensive use of agricultural production are also of great significance to ensure future food production [60]. Moreover, it can also enhance the ability to respond to natural disasters in the future". This clearly indicates that the issue of sustainable agriculture is missed or misunderstood in the agricultural policy. Excessive intensification of production usually leads to environmental degradation, which in the long run is also contrary to the production goals. It would be valuable to discuss this issue before the paper’s publication.
Author Response
Point 1: The amendments done improved the article, but the authors have not yet discussed the issue of sustainability in three parallel dimensions: economic, social and environmental. The authors state that "Promoting intensive agricultural production and improving the level of intensive use of agricultural production are also of great significance to ensure future food production [60]. Moreover, it can also enhance the ability to respond to natural disasters in the future". This clearly indicates that the issue of sustainable agriculture is missed or misunderstood in the agricultural policy. Excessive intensification of production usually leads to environmental degradation, which in the long run is also contrary to the production goals. It would be valuable to discuss this issue before the paper’s publication.
Response 1: Thank you very much for your suggestion and we agree with you. Excessive intensification of production usually leads to environmental degradation, which in the long run is also contrary to the production goals. Therefore, we added the discussions about this issue in the paper. Details are as follows:
Line 640-661
Promoting intensive agricultural production and improving the level of intensive use of agricultural production are also of great significance to ensure future food production[60]. Moreover, it can also enhance the ability to respond to natural disasters in the future.
However, a sustainable food and agriculture system is one which is environmentally sound, economically viable, socially responsible, nonexploitative, and which serves as the foundation for future generations[61-63]. With the long-term development of intensive agriculture production in the Huang Huai Hai Plain, agricultural practices ranging from the development of irrigation projects to the use of agrichemicals have often had negative environmental impacts such as wildlife kills, pesticide residues in drinking water, soil erosion, groundwater depletion, and salinization[64]. Substituting environmentally sound inputs for those which are damaging is an important step in addressing these problems[61]. In view of this, the Ministry of agriculture of China started the construction of Key Laboratory of agricultural environment in Huang Huai Hai Plain with the objectives of scientific research, environmental monitoring, detection analysis and technical services in 2012, aiming to carry out research of regional agricultural pollution prevention by means of agricultural non-point source pollution prevention, environmental protection of producing areas, development and application of environment-friendly inputs. However, for the farmers who are the main body of agricultural production, whether these agricultural technology inputs will increase agricultural production costs and reduce agricultural income will be an important factor affecting the promotion of agricultural technology and the development of sustainable agricultural. Therefore, whether the economic, social and environmental benefits generated by agricultural production in the Huang Huai Hai Plain under the influence of agriculture technology can achieve a balance will be the focus of our future research.
